# Thermophysical Properties of Fe-Si and Cu-Pb Melts and Their Effects on Solidification Related Processes

Rada Novakovic [1],*, Donatella Giuranno [1], Joonho Lee [2], Markus Mohr [3], Simona Delsante [1,4], Gabriella Borzone [1,4], Fabio Miani [5] and Hans-Jörg Fecht [3]

[1] Institute of Condensed Matter Chemistry and Technologies for Energy, National Research Council of Italy (ICMATE-CNR), Via De Marini 6, 16149 Genoa, Italy; donatella.giuranno@ge.icmate.cnr.it (D.G.); simona.delsante@unige.it (S.D.); borzonegabriella@gmail.com (G.B.)

[2] Department of Materials Science & Engineering, Korea University, Anam-dong, Seongbuk-gu, Seoul 136-713, Korea; joonholee@korea.ac.kr

[3] Institute of Functional Nanosystems, Ulm University, Albert-Einstein-Allee 47, 89081 Ulm, Germany; markus.mohr@uni-ulm.de (M.M.); hans.fecht@uni-ulm.de (H.-J.F.)

[4] Department of Chemistry and Industrial Chemistry, Genoa University and Genoa Research Unit of INSTM, Via Dodecaneso 31, 16146 Genoa, Italy

[5] Polytechnic Department of Engineering and Architecture, University of Udine, Via delle Scienze 208, 33100 Udine, Italy; fabio.miani@uniud.it

* Correspondence: rada.novakovic@ge.icmate.cnr.it

**Abstract:** Among thermophysical properties, the surface/interfacial tension, viscosity, and density/molar volume of liquid alloys are the key properties for the modelling of microstructural evolution during solidification. Therefore, only reliable input data can yield accurate predictions preventing the error propagation in numerical simulations of solidification related processes. To this aim, the thermophysical properties of the Fe-Si and Cu-Pb systems were analysed and the connections with the peculiarities of their mixing behaviours are highlighted. Due to experimental difficulties related to reactivity of metallic melts at high temperatures, the measured data are often unreliable or even lacking. The application of containerless processing techniques either leads to a significant improvement of the accuracy or makes the measurement possible at all. On the other side, accurate model predicted property values could be used to compensate for the missing data; otherwise, the experimental data are useful for the validation of theoretical models. The choice of models is particularly important for the surface, transport, and structural properties of liquid alloys representing the two limiting cases of mixing, i.e., ordered and phase separating alloy systems.

**Keywords:** Cu-Pb; Fe-Si alloys; modelling; surface tension; viscosity; molar volume; short-range ordering

## 1. Introduction

An alloy made by mixing and/or fusing two or more metals may have completely different properties with respect to its constituent metals, such as, for example, glass forming ability, a classic example of how properties of alloys can change with respect to those of their pure components [1–5]. The understanding of the nature of metals and how their properties will be changed forming binary or multicomponent alloys is a key issue for optimization of existing alloy systems or the design of new alloys with required properties [6]. During the last thirty years, this has been shown in the cases of lead-free solders [7–9], Ni- and Ti-based superalloys [10–14], biomedical alloys [4,15,16], and so on. The high melting points of some metals, such as Cr, Nb, Mo, Re, W, Ti, Zr, Hf, etc., and their strong chemical reactivity with container materials as well as the affinity for oxygen, in pure state or alloyed, together with the use of inappropriate instrumentation make thermodynamic and thermophysical property measurements difficult or even impossible. Moreover, the experimental data obtained under uncontrolled conditions are often inconsistent, not reproducible, and thus, cannot be

considered as reliable [17]. A reliability improvement can be achieved by applying different experimental techniques for property measurements (a *round robin test*) including the containerless methods using electromagnetic levitation, such as recent experiments done on the International Space Station (ISS) [4,10,11,14,16,18]. Experimental progress made in the last few decades in this field with continuously updating results provide databases with more reliable property data [19] that are requested for microstructure modelling as an integral part of computational materials design [6,13]. The missing data can therefore be estimated based on the model predicted values or by extrapolation from experimentally determined values. Such extrapolations require skills to integrate theory and modelling with experimental observations [13].

The thermophysical properties like density, surface tension, viscosity, diffusivity, thermal expansion, electrical and thermal conductivity, etc., have been widely investigated experimentally and the reference data for almost all pure liquid metals have been assessed [20]. In contrast, although there are many studies on solid alloys and their properties, for the case of liquid alloys, in particular high melting alloy systems, such information is often scarce or not available [4,10,13,14,16]. In spite of the fundamental difference between the amorphous structure of the liquids and the crystalline structure of the solids, the close similarity between the corresponding structures by means of the arrangement of atoms in the two phases is evident at least near their melting temperatures [4,8,11,20,21]. Indeed, the existence of intermetallic compounds with well-defined stoichiometry or miscibility gap in the solid phase and the formation of heterocoordinated n-mers having the same stoichiometry or homocoordinated clusters in the liquid phase, respectively, has been confirmed by diffraction experiments [22,23].

The two limiting cases of atomic interactions observed in liquid Fe-Si and Cu-Pb alloys were analysed in terms of ordering and demixing, indicating structural information related to strongly exothermic [21] and endothermic [24] reactions in these systems, respectively. The Cu-Pb monotectic system is characterized by a miscibility gap in the liquid phase, immiscibility in the solid state, and limited mutual solubility of its pure components [25]. Although both Cu and Pb metals have *fcc* structure, their Goldschmidt atomic radii differ by 37% and thus, they are immiscible in the solid state [24,26]. Since 1958 there have been few assessments of the Cu-Pb phase diagram [25–32]. The monotectic invariant temperature varies between 1225 and 1233 K, whereas the eutectic one is in the range of 599–601 K, as reported in [25–30]. Large positive interaction energies indicate complete demixing in this system and the presence of the two liquid phases in the phase-separated region with Pb-content within 18–67 at % below the critical solution temperature of 1280 K. Above that temperature, the homogeneous liquid phase exists. The enthalpy of mixing, Cu and Pb activities measured at temperature of 1473 K and together with the data on the Gibbs free energy of mixing [33] were taken to calculate the temperature dependent interaction energy and thermodynamic and structural functions of liquid Cu-Pb alloys [34]. On the other side, the calculations of the surface properties of Cu-Pb melts for T = 1373 K were reported in [19]. All thermodynamic datasets of Cu-Pb liquid alloys indicate positive deviation from the Raoult law and together with the Hume–Rothery empirical factors, such as a size ratio ($V_{Pb}/V_{Cu} \approx 2.67$) [20], oxidation state difference (=1; =3) [35], and electronegativity difference after Pauling (=0.1) [36], substantiate the endothermic mixing effects in this system.

The Fe-Si system is a compound forming system characterized by strong interactions between its constituent atoms. Its phase diagram has been assessed by many authors [33,37–39] indicating the formation of FeSi, $Fe_5Si_3$, $Fe_2Si$, $FeSi_2$ and $Fe_3Si_7$ stable intermetallic compounds [37]. The most recent assessments of the Fe-Si phase diagram include the presence of $Fe_2Si$ metastable phase [38,39]. Large negative interaction energies with negative deviations of thermodynamic functions of mixing as well as a size ratio ($V_{Si}/V_{Fe} \approx 1.46$), oxidation state difference (=−1; =−2; =0; =1; =2; =3) [35], and electronegativity difference after Pauling (=0.07) [36] substantiate strong compound forming tendency in the Fe-Si system [21,37–40]. The effects of short-range order on the thermodynamic and

structural properties of Fe-Si melts for T = 1873 K were calculated assuming the prevalence of Fe$_2$Si dimers in the liquid phase and using the compiled thermodynamic datasets [33] as input to the regular associated solution model [40]. Using the Butler model, the surface properties of liquid Fe-Si alloys have been computed for T = 1823 K [19], and the results obtained are close to those obtained by the quasi chemical approximation for regular solution [11].

In order to describe the thermophysical properties relevant for the modelling of solidification of phase separating and strong compound forming alloy systems, such as the Cu-Pb and Fe-Si systems, the most appropriate models were applied. In particular, the model predicted values of the surface and transport properties of similar liquid alloys can differ up to 20%, and therefore, the validation of models using the experimental data is of great importance [4,11,16,18,21,24]. Therefore, the surface properties of the aforementioned systems are described by the self-aggregating model (SAM) [24,41] and compound formation model (CFM) [11,21], respectively. The viscosity of Cu-Pb and Fe-Si melts is analysed by the Moelwyn–Hughes (MH) model [42] and subsequently compared to available literature data. For both systems, the molar volume is calculated using the standard relationships [20] and subsequently compared to the corresponding data obtained from density experimental datasets.

## 2. Theory

### 2.1. Thermodynamics and Surface Properties of Metallic Melts Representing Phase Separation and Strong Compound Forming Tendency

Generally, the mixing behaviour of the constituent atoms classifies liquid binary mixtures into two main groups, i.e., phase separating (demixing) or compound forming (short-range ordered), and within the two groups, the atomic interactions are either strongly repulsive or strongly attractive. There are a limited number of alloy systems exhibiting the characteristics of both groups, for example, Ag-Sb and Ag-Ge [43]. Concerning the limiting cases of mixing, the first one indicates that the attractive forces between similar atoms are much greater than those between dissimilar atoms, and the formation of self-coordinated A-A or B-B pairs takes place leading to demixing and phase separation [24,41]. Demixing and phase separation as its final stage occur due to the formation on homocoordinated clusters, symbolically denoted as

$$i\,A \;\leftrightarrow\; A_i \tag{1}$$

where $i$ is the number of $A$ atoms in an $A_i$ homocoordinated cluster.

In contrast, the second group is characterised by very strong attractive atomic interactions between unlike atoms and the formation of heterocoordinated A-B pairs in a form of $A_\mu B_\nu$ dimers, as follows

$$\mu\,A + \nu\,B \;\leftrightarrow\; A_\mu B_\nu \tag{2}$$

with $\mu$ and $\nu$ stoichiometric coefficients that correspond to those of an energetically favoured intermetallic compound [11,21,44,45].

The case studies regarding liquid Cu-Pb and Fe-Si alloys and their thermodynamic and surface properties have been analysed in the framework of statistical mechanics combined with quasi-lattice theory (QLT) using a unified approach that combines the Bhatia–Thornton and the Singh–Sommer formalisms based on the grand partition function [21,24]. To this aim, the self-aggregating model (SAM) and compound formation model (CFM), respectively, were applied. To evaluate the deviations of the two limiting cases of mixing with respect to the mixing behaviour described by the regular solution model, the quasi chemical approximation for regular solution (QCA) was also used [8,11,41,43–45]. In order to describe the molar volume and viscosity, the thermodynamic models are the most appropriate [20]. All the aforementioned models were validated by means of available experimental datasets. The models used for the property calculations have been described in detail and reported in the literature [11,41,44,45]. In the following, only short descriptions of the models together with the equations used for the property calculations are given.

2.1.1. Phase Separating Liquid Alloys and Self Aggregating Model (SAM)

The thermodynamic properties of monotectic alloy systems characterized by endothermic mixing and phase separation exhibit pronounced positive deviation from the Raoult law, whereas their thermophysical properties show an opposite trend. The presence of homocoordinated clusters (Equation (1)) in the liquid phase, at least near the melting temperature of an alloy, significantly affects the surface properties of monotectic alloys. Therefore, the self-aggregating model (SAM) is the only one that considers clusters of $A$ and $B$ constituent atoms, and it is the most appropriate. Clusters of $A_i$ and $B_j$-type (Equation (1)) are in the form of a polyatomic matrix located on a set of equivalent lattice sites characterised by the interactions of short-range forces that are effective between nearest neighbours only [24,41]. The tendency toward demixing/phase separation depends on the cluster's size and interaction energy between constituent atoms. Under equilibrium conditions between the bulk and surface phases, the surface tension of phase separating alloys using the SAM can be described by

$$\sigma = \sigma_A + \frac{k_B T}{\alpha} \left\{ ln\left(\frac{c_A^s}{c_A}\right) + p\left[ln\left(\frac{c_A \phi^s}{c_A^s \phi}\right) + \frac{(\phi c_A^s - \phi c_A)(\phi^s - \phi)}{c_A c_A^s}\right] + q\left[ln\left(\frac{c_A}{\phi}\right) + \frac{(\phi - c_A)}{c_A}\right] + \frac{iW}{k_B T}\left[p(1-\phi^s)^2 + (q-1)(1-\phi)^2\right] \right\}$$

(3)

or

$$\sigma = \sigma_B + \frac{k_B T}{\alpha} \left\{ ln\left(\frac{c_B^s}{c_B}\right) + p\left[ln\left(\frac{c_B(1-\phi^s)}{c_B^s(1-\phi)}\right) + \frac{(i-j)(\phi^s - \phi)}{i}\right] + q\left[ln\left(\frac{(1-\phi)}{c_B}\right) + \frac{\phi(i-j)}{i}\right] + \frac{jW}{k_B T}\left[p\phi^{s2} + (q-1)\phi^2\right] \right\}$$

(4)

with

$$\phi = \frac{ic_A}{(ic_A + jc_B)}$$

(5)

$$\phi^s = \frac{ic_A^s}{(ic_A^s + jc_B^s)}$$

where $k_B$, $c_A$, $c_B$, $c_A^s$, $c_B^s$ are the Boltzmann constant and the compositions of a bulk and surface phase of an $A - B$ alloy with respect to $A$ and $B$ component; $W$ ($> 0$) is the interaction energy; $p$ and $q$ are the surface coordination numbers; $T$ is temperature; and $i$ and $j$ define the size of $A_i$ and $B_j$-type clusters, respectively [24,41].

2.1.2. Compound Forming Liquid Alloys and Compound Formation Model (CFM)

The surface properties of strongly interacting compound forming systems such as liquid Al-Ni [11], Ag-Hf [44], and Al-Co [45] alloys have been investigated by the CFM that includes strong effects of short-range ordering on these properties. Similar mixing behaviour of liquid Fe-Si alloys implies that the CFM is the most appropriate to describe their thermodynamic and surface properties considering the $AB$-stoichiometry of the *FeSi* intermetallic compound, which is postulated to be energetically favoured [46]. A compound forming binary alloy system with an $A_\mu B_\nu$ energetically favoured intermetallic compound can be considered as a pseudoternary mixture containing $N$ atoms, of which $Nc_A$ and $Nc_B$ are the numbers of $A$ and $B$-atoms. Accordingly, $A$ and $B$-atoms together with $A_\mu B_\nu$ complexes are present in a melt and the three species are in chemical equilibrium. The number of complexes $n_3$ is related to the constituent atoms $A$ and $B$ by

$$n_1 = Nc_A - \mu n_3 \qquad n_2 = Nc_B - \nu n_3 \qquad N = n_1 + n_2 + n_3(\mu + \nu)$$

(6)

with $N$, $\mu$, and $\nu$ denoting Avogadro's number and stoichiometric coefficients of $A$ and $B$ alloy components describing dimer $A_\mu B_\nu$, respectively. In the framework of CFM, the functional form of the Gibbs free energy of mixing $G_M$ is fitted to the experimental data

(the enthalpy of mixing and activities) to obtain four interaction energy parameters [20,45]. Minimizing the Gibbs free energy of mixing for a given temperature and pressure, $n_3$ can be calculated. The last noted together with the interaction energy parameters are necessary to obtain the expressions of the activities of alloy components by means of the standard thermodynamic relations [45], and the surface tension $\sigma$ can be calculated as follows:

$$\sigma = \sigma_A + \frac{k_B T}{\alpha} ln \frac{c_A^s}{c_A} + \frac{k_B T}{\alpha} ln \frac{\gamma_A^s}{\gamma_A} \tag{7}$$

$$\sigma = \sigma_B + \frac{k_B T}{\alpha} ln \frac{c_B^s}{c_B} + \frac{k_B T}{\alpha} ln \frac{\gamma_B^s}{\gamma_B} \tag{8}$$

where $\alpha$, $\sigma_i$, $\gamma_i$, $\gamma_i^s$ $(i = A, B)$ are the mean surface area, surface tensions, activity coefficients of the bulk, and the surface phase of the pure components, respectively [20,45].

### 2.1.3. Quasi Chemical Approximation (QCA) for Regular Solution

The applications of SAM and CFM to describe the surface properties of demixing/phase separating and strongly compound forming alloy systems, which represent the two limiting cases of mixing, highlight the effects of short-range ordering on their surface properties [21,24]. There are two indicators, useful to quantitatively evaluate the contributions of short-range order phenomena on these properties. The first one is deviation from the ideal mixing behaviour, and the second one, more precise, is the use of the quasi chemical approximation (QCA) for regular solution [11,41,44,45] to compare the model predicted property values obtained by the SAM and CFM. Therefore, the difference between the QCA and SAM (or CFM) calculated isotherms indicates the deviation from the regular solution behaviour and is an estimate of the effects of short-range order on the surface properties (surface tension and surface segregation) of the aforementioned groups of alloys [4,8,11,21,24,41,44,45].

The QCA for regular solution is characterised by only one interaction energy parameter that can be obtained from thermodynamic datasets as a function of temperature. In the framework of QCA, the surface tension is calculated by

$$\sigma = \sigma_A + \frac{k_B T(2 - pZ)}{2\alpha} ln \frac{C^s}{C} + \frac{Zk_B T}{2\alpha} \left[ pln \frac{(\beta^s - 1 + 2C^s)(1 + \beta)}{(\beta - 1 + 2C)(1 + \beta^s)} - qln \frac{(\beta - 1 + 2C)}{(1 + \beta)C} \right] \tag{9}$$

$$\sigma = \sigma_B + \frac{k_B T(2 - pZ)}{2\alpha} ln \frac{(1 - C^s)}{(1 - C)} + \frac{Zk_B T}{2\alpha} \left[ pln \frac{(\beta^s + 1 - 2C^s)(1 + \beta)}{(\beta + 1 - 2C)(1 + \beta^s)} - qln \frac{(\beta + 1 - 2C)}{(1 + \beta)(1 - C)} \right] \tag{10}$$

where $Z$ is the coordination number, $\beta$ and $\beta^s$ are composition dependent auxiliary variables for the bulk and surface phases containing the energetic term, and $p$ and $q$ are the surface coordination fractions. For a closed-packed structure, the values of these parameters usually are taken as 1/2 and 1/4, respectively [11,41,45].

### 2.1.4. Perfect Solution Model

Surface tension of alloys exhibiting the ideal behaviour in bulk and in the surface phase can be accounted for by the perfect solution model [47], as follows:

$$exp\left(-\left(\frac{\sigma\alpha}{k_B T}\right)\right) = c_A \cdot exp\left(-\left(\frac{\alpha\sigma_A}{k_B T}\right)\right) + c_B \cdot exp\left(-\left(\frac{\alpha\sigma_B}{k_B T}\right)\right) \tag{11}$$

The variables and constants of Equation (11) are already defined. Otherwise, as noted above, the difference between the most appropriate surface tension isotherm and that calculated by the perfect solution model can be used as one of the indicators for the interactions in an alloy system. Indeed, positive deviations of ideality is related to the systems for which the mixing thermodynamic properties deviate negatively from Raoult's law and vice versa [20].

## 2.2. Transport Properties: Viscosity

The experimental difficulties related to high temperature measurements and/or controversial trends of experimentally determined viscosity datasets are the main problems of development and validation of viscosity models of liquid binary alloys. There are many empirical and semi-empirical viscosity models [20] and after preliminary calculations, the model reported by Moelwyn–Hughes (MH) [42] showed the most appropriate results to predict the viscosity of liquid Cu-Pb and Fe-Si alloys. The thermodynamic MH viscosity model is a very simple one with energetics expressed in terms of the enthalpy of mixing. The MH viscosity isotherm is described by

$$\eta = (c_A \eta_1 + c_B \eta_2)(1 - 2c_A c_B - \frac{H_{mix}}{RT})$$

(12)

where $\eta_1$ and $\eta_2$ are the viscosities of pure components, $H_{mix}$ is the enthalpy of mixing, and $R$ is the gas constant.

## 2.3. Density/Molar Volume

As almost all thermophysical properties relevant for the solidification of, for example, the surface tension, viscosity, molar volume, and compressibility require density data, there is a need to obtain reliable density datasets as input in the models describing the aforementioned issues [10–14]. Until now, there have been no appropriate models to predict the density, and therefore the only sources of such data are the experimental data. The molar volume is the thermophysical property directly related to the density [20]. Indeed, the molar volume of binary alloys in the liquid and the solid state is expressed by

$$V_{Alloy} = \frac{c_A M_A + c_B M_B}{\rho_{Alloy}}$$

(13)

where $M_A$, $M_B$ are the molar masses of components $A$, and $B$ and $\rho_{Alloy}$ is the alloy density. The excess quantities are characteristics of real alloy systems [20,41,43]. Therefore, the excess volume can be calculated by

$$V^E = V_{Alloy} - V_{ideal} = \frac{c_A M_A + c_B M_B}{\rho_{Alloy}} - \left(\frac{c_A M_A}{\rho_A} + \frac{c_B M_B}{\rho_B}\right)$$

(14)

## 2.4. Structural Information: $S_{cc}(0)$ and $\alpha_1$ Microscopic Functions

The determination of the structures of metallic melts by neutron diffraction and X-ray methods is often associated with experimental difficulties related to their high melting temperatures; therefore, there are only a few databases and reviews available [22,23]. The lack of such experimental data can be compensated by the model predicted data that may give information on the nature of mixing and degree of order in a melt in terms of the two microscopic functions, i.e., the concentration-concentration structure factor in the long wavelength limit $S_{cc}(0)$ and Warren–Cowley short-range order parameter $\alpha_1$ (or chemical short-range order parameter, CSRO) [8,21,24,41,43–45]. To this aim, the knowledge of the thermodynamic functions of mixing is required. Knowing the Gibbs energy of mixing $G_M$ of the liquid phase, $S_{cc}(0)$ can be expressed either by $G_M$ or by the activities $a_A$ and $a_B$, as

$$S_{cc}(0) = RT \left(\frac{\partial^2 G_M}{\partial x_A^2}\right)^{-1}_{T,P,N} = x_B a_A \left(\frac{\partial a_A}{\partial x_A}\right)^{-1}_{T,P,N} = x_A a_B \left(\frac{\partial a_B}{\partial x_B}\right)^{-1}_{T,P,N}$$

(15)

For ideal mixing the energy parameters become zero and Equation (12) becomes

$$S_{cc}(0, id) = x_A \cdot x_B$$

(16)

The difference between $S_{cc}(0)$ and $S_{cc}(0, id)$ in terms of inequality involving absolute value defines the mixing behaviour of liquid binary alloys. Indeed, $S_{cc}(0) > S_{cc}(0, id)$ indicates demixing/phase separation, whereas the opposite inequality $S_{cc}(0) < S_{cc}(0, id)$, characterises the presence of chemical order in a melt.

Warren–Cowley short-range order parameter $\alpha_1$ describes the degree of order in liquid alloys and it can be calculated by

$$\frac{S_{cc}(0)}{x_A \cdot x_B} = \frac{1 + \alpha_1}{1 - (Z - 1)\alpha_1} \tag{17}$$

where $\alpha_1$ parameter values are in the range of $-1 \leq \alpha_1 \leq 1$. Its negative values indicate ordering in an alloy melt, and $\alpha_1^{min} = -1$ suggests complete ordering. On the contrary, its positive values indicate demixing, and for $\alpha_1^{max} = 1$, the phase separation takes place.

## 3. Results and Discussion

### 3.1. Thermodynamics of the Two Limiting Cases of Mixing: Cu-Pb and Fe-Si Liquid Alloys

Close similarity of structures between solid and liquid alloys is evident near their melting temperatures. Therefore, to highlight the effects of short-range ordering on the thermodynamic and thermophysical properties of liquid Cu-Pb and Fe-Si alloys, these properties were calculated for T = 1373 and T = 1823 K, near the highest melting temperatures corresponding to Cu and Fe, respectively. Indeed, in the case of alloys exhibiting one of the two limiting cases of mixing, the short-range order effects are evident and can be deduced from the corresponding property curves, their irregularities, and significant deviations from ideality.

### 3.1.1. Cu-Pb

The Cu-Pb phase diagram data [25–33], positive enthalpy of mixing [33,48], as well as the activity datasets [49,50] suggest endothermic mixing behaviour of these alloys, which is substantiated by a monotectic type phase diagram of the Cu-Pb system [25,30–32]. The aforementioned property datasets were used to calculate the interaction energy parameter, which resulted as $W = 1.66\ k_B T$, in agreement with the corresponding data reported in [34]. In the framework of SAM [24,41,51], using the phase diagram data [28–30], the interaction energy term $W$, the enthalpy of mixing [48], Cu [49], and Pb [50] activities, the thermodynamic properties of mixing were calculated for T = 1373 K (Figure 1). The datasets shown in Figure 1 indicate positive enthalpy of mixing and large deviations of the activities from the ideal mixing. A good agreement between the model predicted property values and the corresponding experimental data [48–50] can be observed. Similarly, the calculated values of the Gibbs free energy of liquid Cu-Pb alloys and the phase diagram data [28,29,31] agree fairly well. The tendency towards phase separation and/or demixing in an alloy melt can be "quantified" by the normalised form of the Gibbs free energy of mixing $G_M/RT$ of the liquid phase at the equiatomic composition. For Cu-Pb melts, the values of $G_M/RT = -0.224$ and $H_M/RT = 0.62$ are characteristic for the systems that exhibit demixing and/or phase separation (Figure 1) [24,41,51].

### 3.1.2. Fe-Si

Available experimental data on the thermodynamic properties as well as phase diagram information have been used for the calculation of order energy parameters for the liquid phase of the Fe-Si system. For all measurement temperatures, the Gibbs free energy and the enthalpy of mixing of Fe-Si are negative and exhibit a minimum near the composition $c_{Si} = 0.5$ (Figure 2) and $c_{Si} = 0.46$ (Figure 3), respectively. The thermodynamic studies on the formation of intermetallic compounds in the Fe-Si system suggest retention of order in the melts at equiatomic composition, indicating the $FeSi$ as an energetically favoured compound [46,52]. Accordingly, at least close to the melting temperature, the preferential arrangements of Fe and Si constituent atoms favour the formation of $FeSi$-complexes in the liquid alloys (Equation (2)) with $\mu = 1$ and $\nu = 1$. It is known that the

CFM formalism [11,21,45] considers only the stoichiometry of an energetically favoured compound. Therefore, the asymmetric behaviour of the Fe-Si enthalpy curve and corresponding experimental data [53–55] around the equiatomic composition (Figure 3) may be attributed to the existence of more than one type of association. Ohtani [38] reported $FeSi_2$ having similar formation energy and thus, in the liquid phase, $AB_2$ trimers may be in chemical equilibrium with other chemical complexes and some residual unassociated Fe and Si-atoms present in the melt [11].

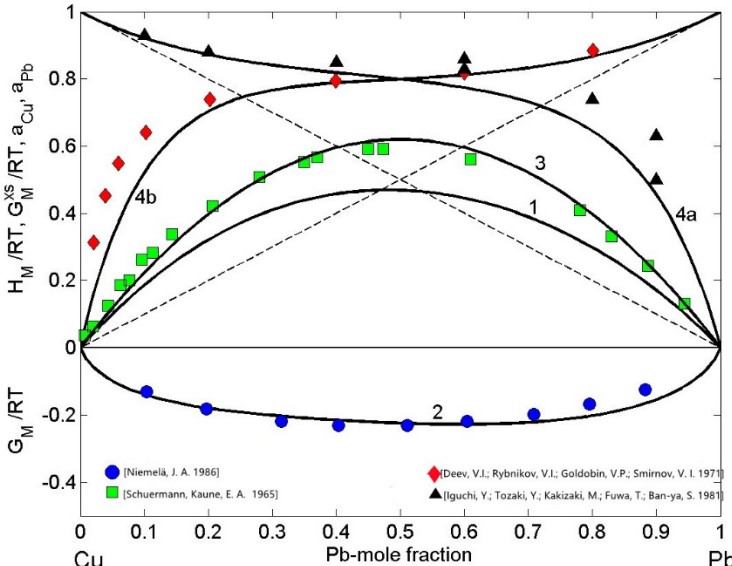

**Figure 1.** Concentration dependence of thermodynamic properties of liquid Cu-Pb alloys calculated for T = 1373 K together with the corresponding experimental data. The excess Gibbs free energy of mixing ($G_M^{xs}$ curve 1); the Gibbs free energy of mixing ($\frac{G_M}{RT}$ [29], curve 2); the enthalpy of mixing ($\frac{H_M}{RT}$ [48], curve 3) and the activities of copper ($a_{Cu}$ [50], curve 4a) and lead ($a_{Pb}$, [55], curve 4b); (- - the ideal mixture).

The datasets of the enthalpy of mixing [53–55], the activities of Si [38,56], and Fe [57] together with the optimised data of the excess Gibbs free energy of mixing of liquid Fe-Si alloys [37] and the Gibbs free energy of mixing [37,39], all obtained at T = 1823 K or close to this temperature, have been used as input data in the CFM to calculate the four interaction energy parameters. The CFM predicted values $g$ = 3.52, $W_{12}$ = −1.55, $W_{13}$ = −1.35, and $W_{23}$ = −1.1, all in $RT$ units, were then used to calculate the concentration dependent number of complexes (Figure 2). As noted above, an indicator for attraction/repulsion between the constituent atoms in an alloy melt is $G_M/RT$, the normalised form of the Gibbs free energy of mixing. The value of $G_M/RT$ = −2.15 (Figure 2) indicates strong atomic interactions and pronounced effects of short-range order in liquid Fe-Si alloys [21,44]. The maximum effect of ordering for T = 1823 K, expressed in $n_3$ of $FeSi$ dimers ($n_3$ = 0.35) at the equiatomic composition, is shown in Figure 2.

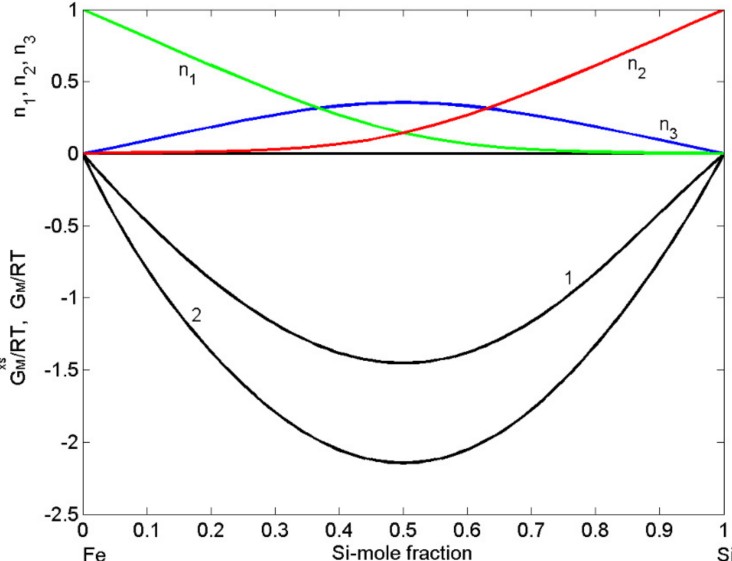

**Figure 2.** Concentration dependence of: the excess Gibbs free energy of mixing ($G_M^{xs}$ curve 1); Gibbs free energy of mixing ($\frac{G_M}{RT}$, curve 2); the equilibrium number of complexes $n_3$ (*FeSi* ) together with unassociated atoms $n_1$ (Fe) and $n_2$ (Si) for liquid Fe-Si alloys calculated by the CFM for T = 1823 K.

The order energy parameters have been used to calculate in the framework of the CFM [45] the enthalpy of mixing, Fe and Si activities of liquid Fe-Si for T = 1823 K. The CFM predicted values of the enthalpy of mixing and the activities agree fairly well with the experimental datasets, and both types of data indicate strong exothermic reactions in this system and large negative deviations from the ideal behaviour (Figure 3). For all calculations, the coordination number Z = 10 was taken [11,45].

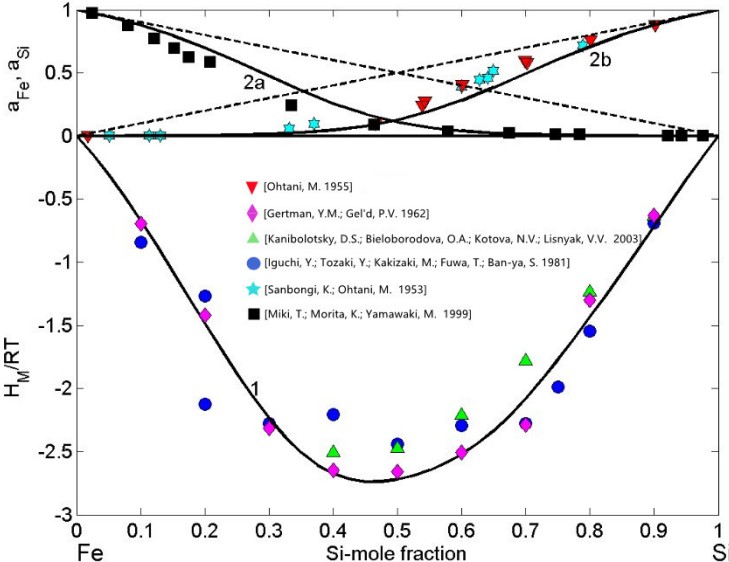

**Figure 3.** Concentration dependence of thermodynamic properties of liquid Fe-Si alloys calculated for T = 1823 K together with the corresponding experimental data: the enthalpy of mixing ($\frac{H_M}{RT}$ [53–55], curve 1); the activities of iron ($a_{Fe}$ [57], curve 2a) and silicon ($a_{Si}$ [38,56], curve 2b). (- - the ideal mixture).

## 3.2. Surface Properties of Phase Separating and Compound Forming Liquid Alloys

The modelling of solidification as the key step of all industrial processes involving the presence of liquid phase, such as casting or joining, requires the knowledge of the surface properties of liquid alloys as input data. To this aim, depending on the mixing behaviour of

liquid alloys, the choice of the most appropriate models validated by the experimental data is the best strategy to select the reliable input data that make possible accurate predictions of solidification, obtaining tailored microstructures [6,10,13].

### 3.2.1. Self Aggregating Model (SAM) and Surface Properties of Cu-Pb Melts

In order to describe the surface tension and surface segregation of liquid Cu-Pb alloys, the three models have been applied. The first one is the SAM [24,51], as the most appropriate for monotectic alloys, followed by the QCA for regular solution [11,41] evaluating the deviations of SAM predicted values from the regular solution surface tension isotherm, and finally, the perfect solution model [11,47], useful for an estimation of the surface properties with respect to the ideal behaviour. In the present work, the surface tension of pure Cu [4] and Pb [58] liquid metals were taken as the reference data together with their molar volume [20], the energetic term $W$, and numbers of atoms (Equation (1)) in self-associates $Cu_i$ and $Pb_j$ obtained from the thermodynamic data [28,29,38,49,50,53–55]. The SAM thermodynamic calculations for T = 1373 K indicate the energetic term value of $W = 1.66 \, k_B T$ and the formation of $Cu_3$ and $Pb_2$ clusters in the melt. Combining Equations (3)–(5), one calculates the surface composition by the SAM, whereas the same property calculated by the QCA for regular solution is obtained subtracting Equation (9) from Equation (10). The calculated values of the surface composition of liquid Cu-Pb alloys suggest the segregation of Pb-atoms to the surface for all bulk concentrations and agree with the fact that the degree of segregation decreases with an increase in temperature (Figure 4). The presence of clusters exhibiting homocoordination tendency increases the segregation on the melt surface and, therefore, the isotherm calculated by the SAM (Figure 4; curve 1) shows higher values with respect to that obtained by the QCA for regular solution (Figure 4; curve 2). The difference between the isotherms (Figure 4; curves 1 and 2) calculated by the two models indicates pronounced effects of clustering on Pb-enrichment in liquid Cu-Pb alloy melts.

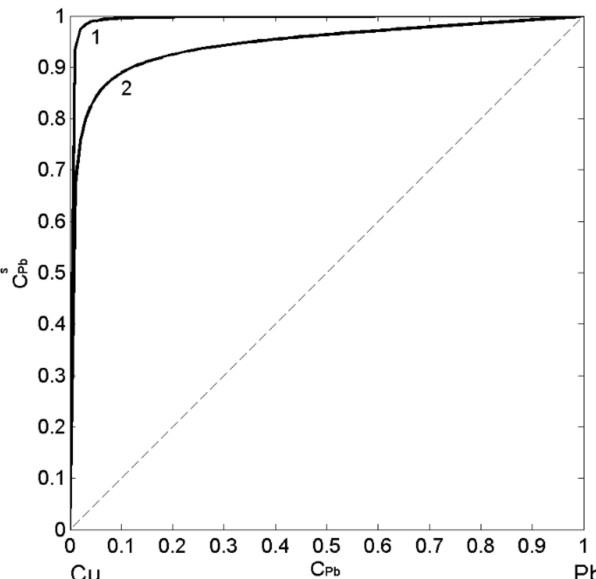

**Figure 4.** Surface composition ($C_{Pb}^S$) vs. bulk composition ($C_{Pb}$) for liquid Cu-Pb alloys calculated by the SAM (curve 1) and the QCA for regular solution (curve 2) for T = 1823 K.

Subsequently, the obtained surface composition values $c_{Pb}^S$ (Figure 4, curve 1) were inserted into either Equation (3) or Equation (4) to calculate the SAM isotherm, whereas in the case of the isotherm calculated by the QCA for regular solution, the corresponding $c_{Pb}^S$ values (Figure 4, curve 2) were inserted in Equation (9) or Equation (10). The surface tension isotherms of liquid Cu-Pb alloys calculated for T = 1373 K together with the literature data [59–61] obtained at the same temperature are shown in Figure 5. The maximum difference between the surface tension data calculated by the SAM (Figure 5, curve 1)

and QCA for regular solution (Figure 5, curve 2) values is approximately 200 $mN/m$ (about 34%) corresponding to $c_{Pb} = 0.2$, that is close to Cu-rich monotectic composition of $c_{Pb} = 0.18 − 0.2$ [28–31]. For the same temperature, the surface tension isotherm of Cu-Pb melt was calculated by the Butler model using the optimized term of the excess Gibbs free energy [28].

The surface tension experimental datasets [59–61] exhibit very good agreement with the isotherm calculated by the SAM and can be used for its validation. On the other side, as expected, very large negative deviations from the ideal behaviour (Equation (11)) described by the perfect solution isotherm (Figure 5, curve 3) can be observed [47]. The surface tension isotherms of Cu-Pb melts, calculated by the SAM and QCA for regular solution, deviate negatively with respect to that calculated by the perfect solution model (Figure 5), confirming that liquid alloys with positive deviations of mixing properties (Figure 1) exhibit negative deviations of their thermophysical properties [20]. Concerning clustering effects on the surface segregation and surface tension, it is important to note that such effects are reciprocal, as can be seen in Figures 4 and 5. Indeed, the surface tension isotherm obtained by the SAM (Figure 5, curve 1) is lower than that calculated by the QCA for the regular solutions, in agreement with the previous considerations related to the segregation of Pb-atoms on the surface of Cu-Pb melts (Figure 4, curve 1).

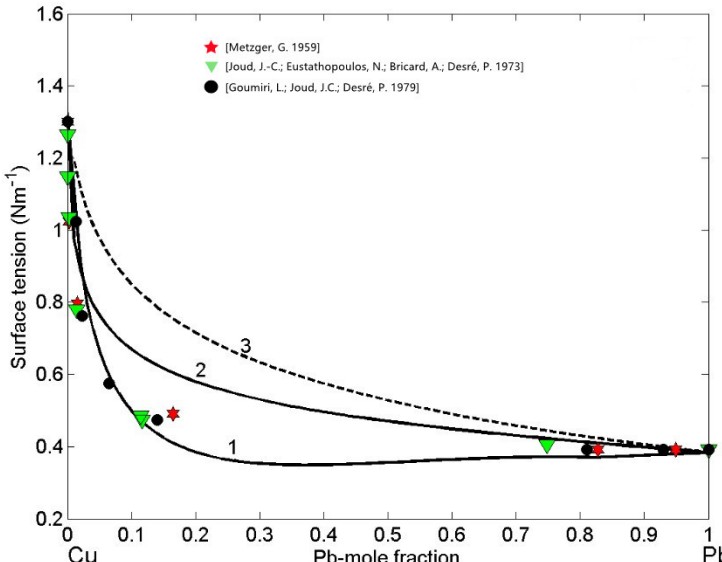

**Figure 5.** Surface tension isotherms of liquid Cu-Pb alloys calculated by: the SAM (curve 1), the QCA for regular solution (curve 2), and the perfect solution model (curve 3) for T = 1373 K. For a comparison, the available experimental data [59–61] obtained at the same temperature are shown.

3.2.2. Compound Forming Model (CFM) and Surface Properties of Fe-Si Melts

Using the thermodynamic data [37,38,53–57] related to mixing behaviour of liquid Fe-Si alloys, their surface properties as functions of composition have been calculated for T = 1823 K by the two models: the CFM and the QCA for regular solution. Postulating the *FeSi* intermetallic compound as energetically favoured (Equation (2)) [46,52], it is assumed the formation of *FeSi* dimers ($\mu = 1\ v = 1$) in the liquid phase, at least near the melting temperature and solving Equations (7) and (8), the CFM surface tension isotherm was calculated. Similarly, using Equations (9) and (10), the QCA isotherm for regular solution was obtained (Figure 6; curve 2). The calculated values of surface composition $c_{Si}^{S}$ suggest the enrichment by Si-atoms over the whole composition range (Figure 6; curve 1 and curve 2) and agree that the degree of segregation decreases with an increase in temperature. The pronounced effects of the short-range order in Fe-Si melts suppress Si-segregation on the surface layer (Figure 6, curve 1). The magnitude of these effects can be estimated by the difference in the surface composition calculated by the CFM and QCA for regular

solution model. In particular, the maximum difference in Si-segregation, indicated by the two calculated isotherms, is evident for $c_{Si} < 0.5$, suggesting the concomitant effects of *FeSi* dimers and associates with stoichiometry of other Fe-rich intermetallics (Figure 6).

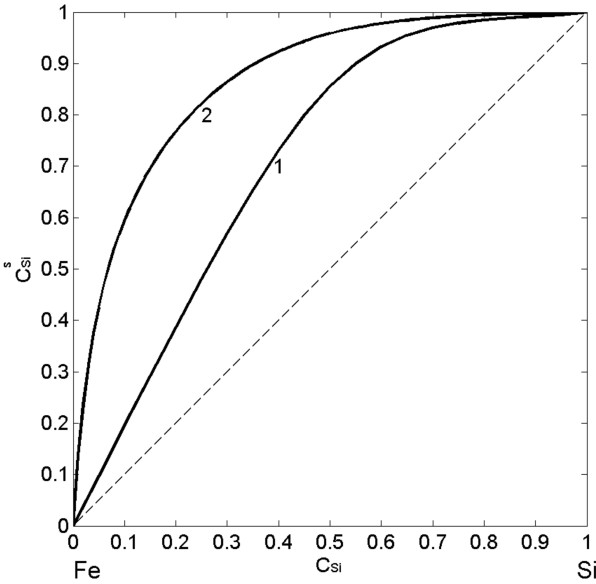

**Figure 6.** Surface composition ($C_{Si}^s$) vs. bulk composition ($C_{Si}$) for liquid Fe-Si alloys calculated by the CFM (curve 1) and the QCA for regular solution (curve 2) for T = 1823 K.

The thermodynamic datasets of liquid Fe-Si alloys [37–39,53–57], the surface tension reference data of liquid Fe [62] and Si [63], and their the molar volumes data [20] have been taken as the input data to calculate the surface tension isotherms. Using the CFM (Equations (7) and (8)) and the QCA for regular solution (Equations (9) and (10)) as well as the perfect solution model, the calculations were performed for T = 1823 K and compared to the literature experimental data [18,64–69]. Large differences between the model predicted values by the CFM (Figure 7, curve 1) and QCA for regular solution (Figure 7, curve 2) indicate strong compound forming tendency in liquid Fe-Si alloys [21]. The surface tension isotherms of liquid Fe-Si alloys obtained by the CFM and QCA for regular solution deviate positively with respect to that calculated by the perfect solution model, confirming that liquid alloys with negative mixing thermodynamic properties in the bulk exhibit positive surface tension deviations with respect to the ideal behaviour (Figure 7, curve 3). In all calculations, for the Fe-Si liquid phase, the coordination number Z = 10 was taken [4,11,21]. The experimental data [18,64–69] exhibit excellent agreement with the CFM isotherm within $0.15 < c_{Si} < 0.5$, otherwise the data agree fairly well with the QCA model predicted values. Among the datasets noted above, it is important to note the most recent surface tension data of Fe-10 wt% Si alloy (Fe-18.1 at %) described by ($\sigma = 1.656 - 0.359 \cdot 10^{-4}(T - 1774 \ K)$) [18], obtained by short-time measurements on parabolic flights thanks to containerless levitation processing [10,11]. Slightly lower surface tension data [64,66] were observed for Si-rich alloys that are more prone to oxidation with respect to Fe-Si alloys containing less silicon. The effects of trace quantity of surfactants in the surrounding atmosphere, mainly oxygen and sulphur, that seem to be insignificant for most thermodynamic measurements, significantly reduce the surface properties of molten metals and alloys, such as those of liquid Si, Fe, and their alloys [20]. Generally, the concomitant effects of oxidation and the presence of two or more intermetallic compounds with similar energetics in an alloy melt, favour complex effects of the short-range order phenomena and, to predict in a rigorous manner the surface tension behaviour, new models are needed.

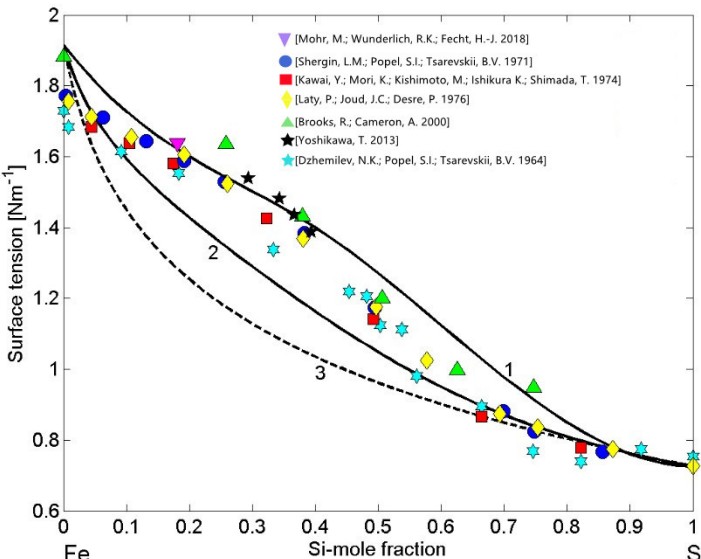

**Figure 7.** Surface tension isotherms of liquid Fe-Si alloys calculated by: the CFM (curve 1), the QCA for regular solution (curve 2), and the perfect solution model (curve 3) for T = 1823 K. For a comparison, the available experimental data [18,64–69] obtained at the same temperature are shown.

### 3.3. Molar Volume/Density of Cu-Pb Phase Separating and Fe-Si Compound Forming Liquid Alloys

The density of liquid Cu-Pb and Fe-Si alloys is of great importance for casting processes of Cu- and Fe-based alloys, including steels. In addition, the density of Fe-Si alloys plays important role in smelting reduction processes [70]. Therefore, concerning the mixing behaviour of metallic melts near their melting temperatures, the molar volume as the thermophysical property derived from density is more useful. It displays a step-like behaviour on melting, indicating a first-order transition, or it links qualitatively the changes in volume with those in thermodynamic mixing functions and structural ordering on the short-range scale [20]. The analysis of the molar volume data of phase separating alloys [71,72] indicates that the excess volume is close to zero, as also observed in the case of the molar volume of liquid Cu-Pb alloys exhibiting near ideal behaviour [73]. This can be explained by the strong demixing tendency that drives phase separation in the miscibility gap resulting in the high positive repulsive forces between Cu and Pb [24]. Using the reference data for the molar volume of liquid Cu [74] and Pb [75], the calculated molar volume of liquid Cu-Pb alloys (Equations (13) and (14)) together with the experimental data [73] obtained at T = 1373 K are shown in Figure 8. The molar volume isotherm (Figure 8, curve 1) of liquid Cu-Pb alloys exhibits very small deviations from the ideal mixture (Figure 8, curve 2) in agreement with the experimental dataset [73].

The molar volume reference data of liquid Fe [76], Si [77], and Fe-Si alloys [18,53,68,69,78] were used to calculate the molar volume isotherm (Equations (13) and (14)) for T = 1823 K (Figure 9, curve 1). The high negative attractive forces between Fe atoms and Si atoms indicate pronounced short-range ordering in the melts and, therefore, the molar volume isotherm of liquid Fe-Si alloys (Figure 9, curve 1) exhibits large negative deviations from the ideal mixture (Figure 9, curve 2), as was already observed in the case of Al-Ni melts [79]. All the experimental datasets of the molar volume [18,53,68,69,78] follow the same trend and, with the exception of dataset reported in [69], they are close to the isotherm (Figure 9, curve 1). In particular, the molar volume data [18] was deduced from the density (Equation (13)) that was previously determined from the sample radius and the sample mass of the samples processed under reduced gravity conditions with an electromagnetic processing device on board a parabolic flight airplane [18].

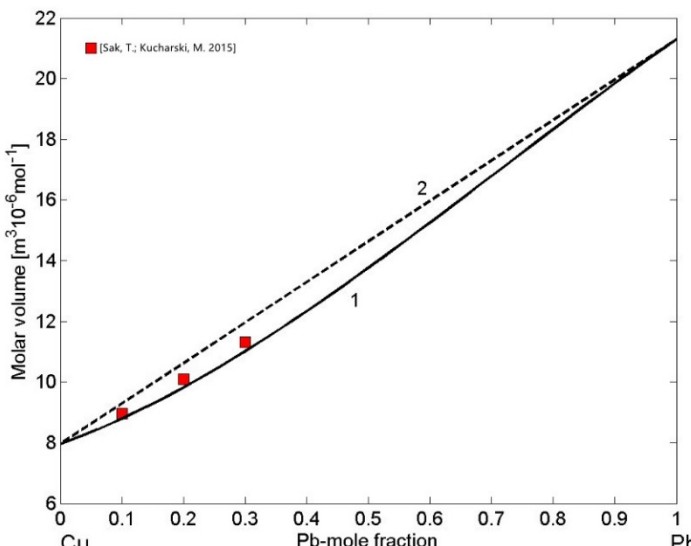

**Figure 8.** Concentration dependence of the molar volume (curve 1) and the ideal mixture (curve 2) of liquid Cu-Pb alloys calculated for T = 1373 K. For a comparison, the molar volume values obtained from the density data [73] measured at the same temperature are shown.

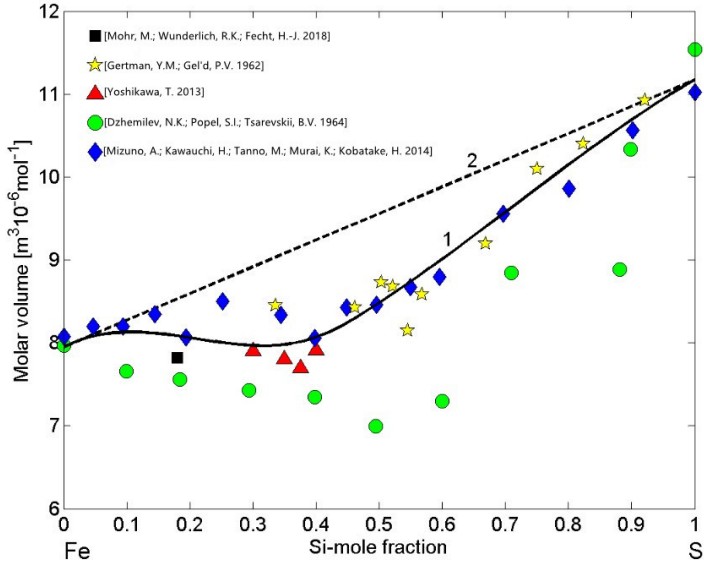

**Figure 9.** Concentration dependence of the molar volume (curve 1) and the ideal mixture (curve 2) of liquid Fe-Si alloys calculated for T = 1823 K. For a comparison, the molar volume values obtained from the density data [18,53,68,69,78] measured at the same temperature are shown.

*3.4. Viscosity of Cu-Pb Phase Separating and Fe-Si Compound Forming Liquid Alloys*

Preliminary analysis of the viscosity models reported in [20] and their applications to liquid Cu-Pb alloys have been performed by Terzieff [80], suggesting his four parameters model [81] as the most appropriate. Terzieff's model is an extension of Morita's model [20], and both models are based on the MH model [42] and include hard sphere and four empirical parameters contributions. Terzieff's viscosity isotherm has been calculated for T = 1373 K using the enthalpy of mixing of liquid Cu-Pb [33] compiled for T = 1473 K together with the Cu and Pb viscosity reference data [82] measured at T = 1373 K. It has similar shape to that of the MH model (Equation (12)) and agrees well with the experimental dataset [82]. Therefore, in the present work, to describe the viscosity of liquid Cu-Pb alloys, the MH model [42] with the enthalpy of mixing [48] obtained at T = 1373 K was used and, subsequently, this model was validated by the datasets [82–84]. To this aim, using

other viscosity reference data as input, the two isotherms were calculated for T = 1373 K. In the first case, the most recent viscosity data of pure Cu [74] and Pb [85] liquid metals were taken as the reference data to calculate the Cu-Pb isotherm (Figure 10, curve 1a), whereas the second viscosity isotherm (Figure 10, curve 2a), was obtained using only the experimental dataset [82] that also includes the viscosity of the pure alloy components. The last one clearly indicates that the MH model [42] is appropriate for Cu-Pb alloys. Subsequently, the model-predicted viscosity values of Cu-Pb melts (Figure 10, curve 1a) were compared with all available datasets [82–84]. The two MH viscosity isotherms (Figure 10, curves 1a and curve 2a) and the datasets [82,84] deviate negatively from the ideal solution isotherms (Figure 10, curves 1b and 2b). The kinematic viscosity datasets [83,84] have been measured on heating (h) and on cooling (c) and, using the extrapolation of composition dependent Cu-Pb density data [73], those data were transformed into dynamic viscosity. The datasets [83,84] obtained on cooling (c) agree fairly well with the viscosity isotherm (Figure 10, curve 1a), whereas those measured in heating (h) [83] exhibit larger deviations.

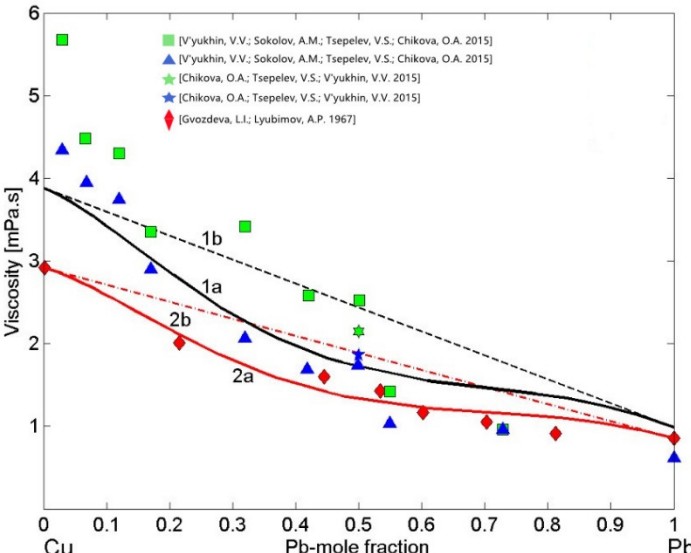

**Figure 10.** Viscosity isotherms of liquid Cu-Pb alloys calculated by Moelwyn–Hughes's (MH) model for T = 1373 K using different viscosity reference data of pure liquid metals: 1—the viscosity isotherms with the recommended data of Cu [74] and Pb [85] (curve 1a and curve 1b); 2—the viscosity isotherms with experimental data [82] (curve 2a and curve 2b). For a comparison, the experimental data [82–84] are shown (h—heating, c—cooling; curves 1b and 2b—the ideal mixture).

Theoretical analysis of four viscosity models applied to liquid Fe-Si alloys for T = 1873 K [86] indicated Kaptay's [87] model as the most appropriate. In contrast, considering the viscosity dataset [88] of liquid Fe-Si alloys as well as those of Al-Co [45] or Al-Ni melts [89], one may expect an irregular viscosity isotherm. It is well known that the properties curves, such as the viscosity, surface tension, molar volume, electrical resistivity, and other isotherms of strongly interacting compound forming systems exhibit a pronounced irregularity over the composition range characterised by the presence of few intermetallic compounds in the solid state [21]. In the liquid phase, over the aforementioned composition range, pronounced effects of short-range order can be observed. Therefore, to describe the viscosity of liquid Fe-Si alloys, the simple thermodynamic MH model [42] may be the most appropriate. To this aim, the viscosity of liquid Fe [76] and Si [77] were taken as the reference data and combined with the enthalpy of mixing [53] into Equation (12). It is important to note that recently measured viscosities of the pure alloy components [76,77] are close to those reported in [85]. The viscosity isotherm (Figure 11, curve 1) calculated for T = 1823 K exhibits positive deviation from the ideal behaviour (Figure 11, curve 2).

Concerning the experimental datasets of Fe-Si melts (Figure 11), one can distinguish two groups of data: the first group of datasets [18,90,91] are close to the ideal viscosity isotherm (Figure 11, curve 2). Among them, the data obtained by the oscillating drop method in an electromagnetic levitation on board of parabolic flights [18] is slightly above the corresponding ideal value (Figure 11). The second group is dataset [88], significantly higher with respect to the viscosity isotherm (Figure 11, curve 1). Indeed, the kinematic viscosity dataset [88] was transformed into dynamic viscosity using the density data obtained by the same authors. The experimental dataset [88] follows the trend of the MH viscosity isotherm, but differs significantly from the model predicted values. Usually, large differences between the kinematic viscosity data and the corresponding theoretical values may be attributed to the errors of measured viscosity or density data, keeping in mind that the viscosity of oxidised melts increases.

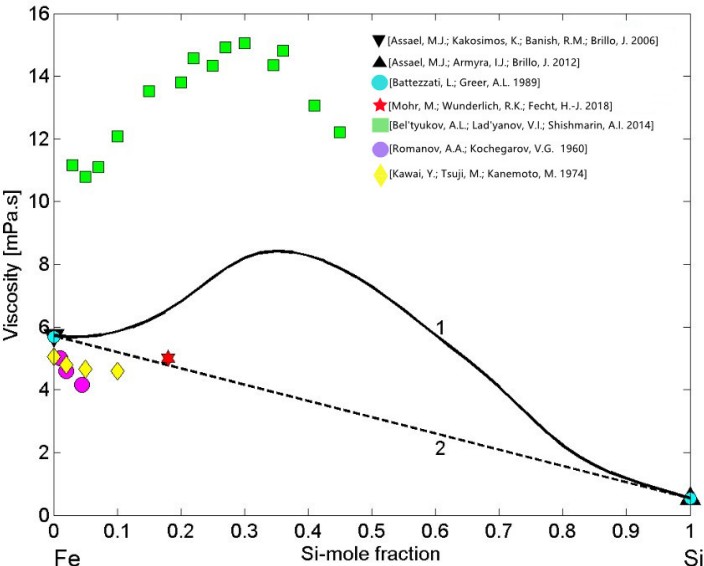

**Figure 11.** Viscosity isotherms of liquid Fe-Si alloys calculated by Moelwyn–Hughes's (MH) model (curve 1) for T = 1823 K. The viscosity reference data of liquid Fe [76,85] and Si [77,85] together with available experimental datasets of Fe-Si melts [18,88,90,91] are shown; the ideal mixture (curve 2).

*3.5. Microscopic Functions of Cu-Pb Phase Separating and Fe-Si Compound Forming Liquid Alloys*

The ordering phenomena in liquid Cu-Pb alloys have been analysed in terms of the concentration fluctuations in the long wavelength limit $S_{cc}(0)$ (Equations (12) and (13)) and CSRO parameter $\alpha_1$ (Equation (14)) as functions of bulk composition for T = 1373 K (Figure 12). In order to visualize the nature of atomic interactions in an alloy melt, $(S_{cc}(0) - S_{cc}(0, id))$ is the indicator. In the case of Cu-Pb melts, $S_{cc}(0) > S_{cc}(0, id)$ (Figure 12, curves 1 and 3) indicates demixing and phase separation. Approaching the monotectic line, the temperature dependence of limit $S_{cc}(0)$ is very sensitive and it tends to infinity. Indeed, in the region of the miscibility gap, $S_{cc}(0)$ increases sharply with an increase in temperature. Accordingly, at the monotectic temperature T = 1228.5 K of the Cu-Pb system [31], $S_{cc}(0)$ tends sharply to infinity, and with an increase in temperature up to T=1373 K, $S_{cc}(0)$ decreases reaching the maximum value of 2.12 for to $c_{Pb} = 0.4$ (Figure 12, curve 1). Similarly, positive values of the Warren–Cowley short-range order parameter (Figure 12, curve 2) over the whole composition range with the maximum value of $\alpha_1 = 0.077$ for $c_{Pb} = 0.5$ display strong tendency towards homocoordination forming $Cu_3$ (Cu-Cu) and $Pb_2$ (Pb-Pb) clusters as nearest neighbours. Both microscopic functions show maximum demixing leading to phase separation for compositions in the range of $0.4 < c_{Pb} < 0.5$ (Figure 12) that are close to the monotectic compositions of $c_{Pb} = 0.18$ and $c_{Pb} = 0.57$ [31]. The CSRO parameter $\alpha_1$ of the two Cu-35Pb and Cu-65Pb (in at %) alloys was

obtained for neutron diffraction experimental data [22] and exhibit a good agreement with the corresponding calculated values (Figure 12, curve 2).

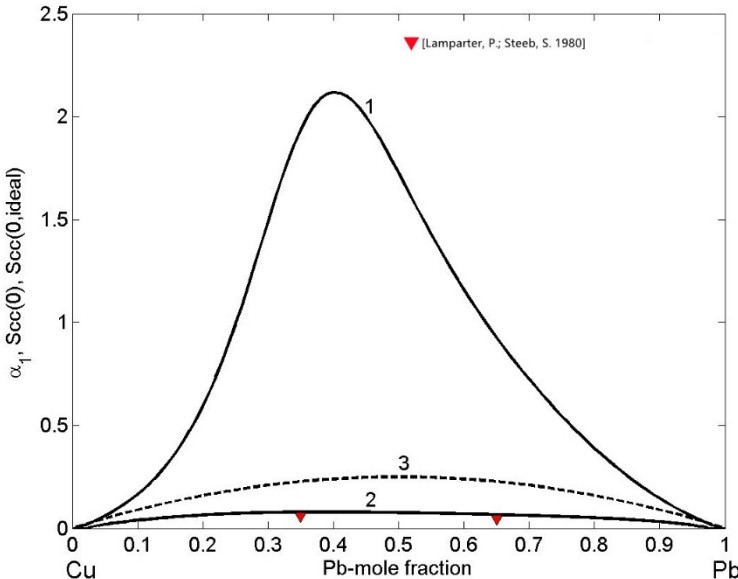

**Figure 12.** Composition dependent concentration fluctuations in the long-wavelength limit $S_{cc}(0)$ (curve 1), $S_{cc}(0, id)$ for the ideal mixing (curve 3) and chemical short-range order parameter $\alpha_1$ (curve 2) of liquid Cu-Pb alloys calculated for T = 1373 K. For a comparison with $\alpha_1$, the experimental data [22] are shown.

The structural properties of liquid Fe-Si alloys, expressed in terms of the microscopic functions, i.e., the concentration fluctuations in the long-wavelength limit $S_{cc}(0)$ (Equations (12) and (13)) and the CSRO parameter $\alpha_1$ (Equation (14)), as functions of bulk composition indicate complex formation tendency in Fe-Si melts and the importance of the *FeSi* energetically favoured intermetallic compound [46,52]. Indeed, $S_{cc}(0)$ values clearly show $S_{cc}(0) < S_{cc}(0, id)$ (Figure 13, curve 1 and curve 3) over the whole composition range, and, together with negative values of CSRO parameter $\alpha_1$ (Figure 13, curve 3), support strong compound forming tendency in Fe-Si melts. The largest difference between $S_{cc}(0)$ and $S_{cc}(0, id)$ is within the composition interval of $0.11 < C_{Si} < 0.73$ that is according to the Fe-Si phase diagram [37,39], characterised by the presence of a few intermetallic compounds and the formation of heterocoordinated associates in the liquid phase, at least close to the melting temperatures of alloys [11,21,45]. *FeSi* associates are prevalently present in Fe-Si melts, but also the associates with other stoichiometries, corresponding to those of other intermetallic compounds, may be present [22,23]. On the other side, the shape and relatively law values of $S_{cc}(0)$ with a slightly lower flat minimum of 0.054 at $C_{Si} = 0.45$–0.46 together with a symmetric curve of the CSRO parameter $\alpha_1$ with respect to $C_{Si} = 0.52$, near the equiatomic composition and its deep minimum of $-0.23$, substantiate the presence of *FeSi* associates, indicating strong effects of short-range ordering phenomena in liquid Fe-Si alloys.

Other important information concerns the glass forming ability of the Fe-Si system. It can be deduced from the shape of composition dependent $S_{cc}(0)$ and $S_{cc}(0, id)$ curves (Figure 13, curve 1 and curve 3). Indeed, for Fe-rich alloys containing up to 89 at % Fe, the two curves $S_{cc}(0)$ and $S_{cc}(0, id)$ overlap, showing that the difference $|S_{cc}(0) - S_{cc}(0, id)| = 0$ may be considered as a necessary condition for glass formation. A random mixing between $A_\mu B_\nu$ associates, A and B unassociated atoms can presumably hinder the nucleation of new phases, but the ability of liquid alloys to reach the glassy state also involves many other kinetic factors [1–3]. Some examples of Fe-Si based bulk metallic glasses (BMG) are ternary systems such as Fe-Si-B, Fe-Si-Al, and Fe-Si-Ti [92].

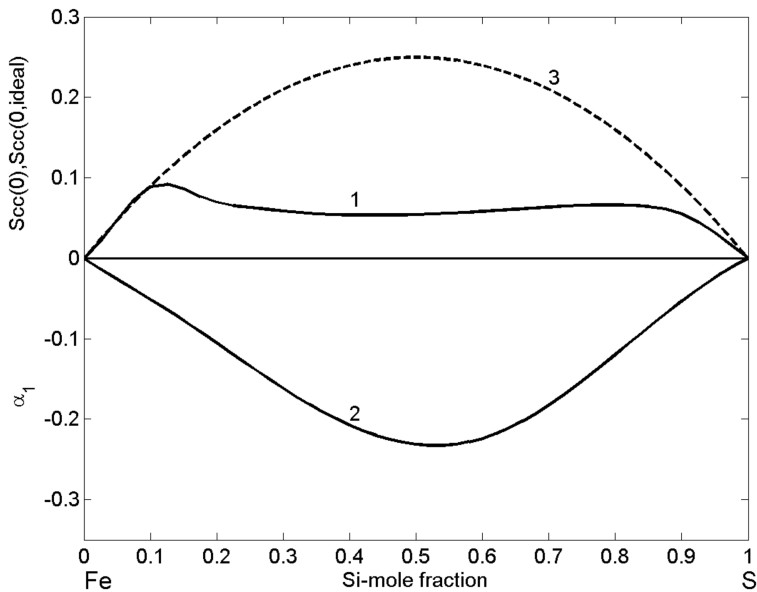

**Figure 13.** Composition dependent concentration fluctuations in the long-wavelength limit $S_{cc}(0)$ (curve 1), $S_{cc}(0, id)$ for the ideal mixing (curve 3), and chemical short-range order parameter $\alpha_1$ (curve 2) of liquid Fe-Si alloys calculated for T = 1823 K.

## 4. Conclusions

Thermophysical properties data of liquid Cu-Pb and Fe-Si alloys are used as input data for numerical simulations of solidification as a step of all industrial processes that involve the presence of the liquid phase, such as casting and joining. In particular, the production of structural components/parts by casting and quality of the products depends on the microstructural evolution and possible formation of defects, which on the other side, are directly related to numerical simulation of solidification. Surface tension, density/molar volume, and viscosity datasets of liquid Cu-Pb and Fe-Si alloys have been mainly obtained by container-based methods, whereas only one dataset related to the Fe-10 wt% Si alloy is the result of containerless measurements. Indeed, the surface tension and viscosity of the Fe-10 wt% Si were measured by the oscillating drop method in an electromagnetic levitation device on board a parabolic flight airplane. The experimental data obtained under microgravity conditions make possible accurate measurements at high temperatures. Therefore, the accuracy and reliability as well as the storage of the thermophysical properties datasets are the main issue for the modelling of solidification related industrial processes. The thermophysical properties datasets of liquid Cu-Pb and Fe-Si alloys were used to validate the predictive models, indicating that only the synergy between the experimental and theoretical data can result in successful materials and processes design.

**Author Contributions:** Conceptualization, R.N., D.G., J.L., S.D., F.M. and H.-J.F.; data curation, D.G., M.M., and S.D.; formal analysis, D.G., S.D. and F.M.; funding acquisition, R.N., D.G., M.M. and H.-J.F.; investigation, R.N., D.G., J.L., M.M. and S.D.; methodology, R.N., D.G., J.L. and H.-J.F.; resources, R.N.; software, R.N.; supervision, J.L., G.B. and H.-J.F.; validation, R.N., D.G., J.L. and M.M.; writing—original draft, R.N., D.G., M.M., S.D., G.B. and F.M.; writing—review and editing, D.G., S.D. and G.B. All authors have read and agreed to the published version of the manuscript.

**Funding:** This research received no external funding.

**Institutional Review Board Statement:** The study did not require ethical approval.

**Informed Consent Statement:** Not applicable.

**Data Availability Statement:** Not applicable.

**Acknowledgments:** The parabolic flight was supported by German Space Agency, DLR, and by the European Space Agency, ESA. The authors are grateful for the support by the team of the Institut

für Materialphysik im Weltraum at DLR Cologne for conducting the parabolic flight experiments with the electromagnetic levitation device. Support for the project Thermolab by the German Space Agency under Contract 50WM1759 and by the ESA Microgravity Support Programme under Contract AO-2009-1020 is gratefully acknowledged. The authors also acknowledge Begona Santillana, TATA Steel Europe, and Prakash Srirangam, University of Warwick, for their support and for providing material samples.

**Conflicts of Interest:** The authors declare no conflict of interest.

## Abbreviations

| | |
|---|---|
| $A$, $B$ | components of an $A - B$ alloy |
| $a_i$ ($i = A, B$) | activity of component $i$ |
| $c_i$ ($i = A, B$) | composition of component $i$ |
| $c_i^s$ ($i = A, B$) | surface composition of component $i$ |
| $g$ | energetic term of CFM |
| $G_M$ | Gibbs free energy of mixing |
| $G_M^{xs}$ | excess Gibbs free energy of mixing |
| $H_{mix}$ | enthalpy of mixing |
| $k_B$ | Boltzmann's constant |
| $M_i$ ($i = A, B$) | atomic mass of component $i$ |
| $n_i$ ($i = 1, 2, 3$) | number of specie $i$ according to CFM in an $A - B$ alloy |
| $N$ | Avogadro's number |
| $p, q$ | surface coordination fractions |
| $R$ | gas constant |
| $S$ | surface area of an alloy |
| $S_{cc}(0)$ | concentration fluctuations in the long wavelength limit |
| $S_{cc}(0, id)$ | concentration fluctuations for the ideal mixing |
| $T$ | absolute temperature |
| $V_i$ ($i = A, B$) | atomic volume of the component $i$ |
| $V^E$ | excess volume |
| $V_{Alloy}$ | volume of a liquid $A - B$ alloy |
| $Z$ | coordination number |
| $W$ | interaction energy term of SAM |
| $W_i$ ($i = 1, 2, 3$) | energetic terms of CFM |
| $\alpha$ | mean surface area of an $A - B$ alloy |
| $\alpha_i$ ($i = A, B$) | surface area of atomic species $i$ |
| $\alpha_1$ | short-range order parameter |
| $\beta$ | auxiliary function for the bulk phase |
| $\beta^s$ | auxiliary function for the surface phase |
| $\gamma_i$ ($i = A, B$) | activity coefficient of component $i$ |
| $\phi, \phi^s$ | composition functions of the bulk and surface phases |
| $\eta$ | viscosity of $A - B$ liquid alloys |
| $\eta_i$ ($i = A, B$) | viscosity of component $i$ |
| $\mu, \nu$ | stoichiometric coefficients of an intermetallic |
| $\rho_i$ ($i = A, B$) | density of component |
| $\rho_{Alloy}$ | density of a liquid $A - B$ alloy |
| $\sigma$ | surface tension of liquid $A - B$ alloys |
| $\sigma_A$ | surface tension of pure component $A$ |
| $\sigma_B$ | surface tension of pure component $B$ |

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
