# Peer review of "Thermophysical Properties of Fe-Si and Cu-Pb Melts and Their Effects on Solidification Related Processes"

_metals, doi:10.3390/met12020336_

Round 1

Reviewer 1 Report

In this manuscript, the authors systematacially analysized the thermophysical properties of the Fe-Si and Cu-Pb systems by numerical simulations of solidification.

I recommend to publish the manuscript in the present form. The methods are clearly described and the manuscript is well structured. The motivation for this research is justified, the methods are perfectly suited.

Author Response

Dear Reviewer 1, 

Thank you very much for dedicating your time to review our manuscript.

On behalf of myself and the co-authors

Best regards

Rada Novakovic

Reviewer 2 Report

This work discusses the importance of accurate thermosphysical properties. The property values obtained from different methods (calculation, experiment and literature data) are compared. The conclusion seems reasonable. To be more attractive, the abstract could be rephrased by focusing on the contribution of the present work rather than its significance. As a whole, publication is fine with me.

Author Response

Dear Reviewer 2, 

On behalf of myself (the corresponding author) and the coauthors, we thank you to dedicate your time to review our manuscript. As suggested, we shifted last phrase relating to the systems investigated, inserting it into the upper part of the Abstract. In this way we focused on the contribution of the present work, i.e. the thermophysical properties of liquid Fe-Si and Cu-Pb alloys. In the revised version of the manuscript, all changes are indicated "in red".

Best regards

Rada Novakovic

Abstract: Among the thermophysical properties, the surface / interfacial tension, viscosity and density / molar volume of liquid alloys are the key properties for the modelling of microstructural evolution during solidification. Therefore, only reliable input data can yield accurate predictions preventing the error propagation in numerical simulations of solidification related processes. To this aim, the thermophysical properties of the Fe-Si and Cu-Pb systems were analysed and the connections with the peculiarities of their mixing behaviours are highlighted. Due to experimental difficulties related to reactivity of metallic melts at high temperatures, the measured data are often unreliable or even lacking. The application of containerless processing techniques either leads to a significant improvement of the accuracy or makes the measurement possible at all. On the other side, accurate model predicted property values could be used to compensate the missing data; otherwise, the experimental data are useful for the validation of theoretical models. The choice of models is particularly important for the surface, transport and structural properties of liquid alloys representing the two limiting cases of mixing, i.e. ordered and phase separating alloy systems.

Reviewer 3 Report

The article can be published in presented form.

Author Response

Dear Reviewer 3

Thank you very much for dedicating your time to review our manuscript.

On behalf of myself and the coauthors

Best regards

Rada Novakovic